# Light Up Your Face: A Physically Consistent Dataset and Diffusion Model for Face Fill-Light Enhancement

Jue Gong [* 1]   Zihan Zhou [* 1]   Jingkai Wang [1]   Xiaohong Liu [1]   Yulun Zhang [† 1]   Xiaokang Yang [1]

## Abstract

Face fill-light enhancement (FFE) brightens underexposed faces by adding virtual fill light while keeping the original scene illumination and background unchanged. Most face relighting methods aim to reshape overall lighting, which can suppress the input illumination or modify the entire scene, leading to foreground–background inconsistency and mismatching practical FFE needs. To support scalable learning, we introduce LightYourFace-160K (LYF-160K), a large-scale paired dataset built with a physically consistent renderer that injects a disk-shaped area fill light controlled by six disentangled factors, producing 160K before-and-after pairs. We first pretrain a physics-aware lighting prompt (PALP) that embeds the 6D parameters into conditioning tokens, using an auxiliary planar-light reconstruction objective. Building on a pretrained diffusion backbone, we then train a fill-light diffusion (FiLitDiff), an efficient one-step model conditioned on physically grounded lighting codes, enabling controllable and high-fidelity fill lighting at low computational cost. Experiments on held-out paired sets demonstrate strong perceptual quality and competitive full-reference metrics, while better preserving background illumination. The dataset and model will be at https://github.com/gobunu/Light-Up-Your-Face.

## 1. Introduction

Face fill-light enhancement (FFE) is a specific form of face relighting that targets facial images that are underexposed or captured under suboptimal lighting setups, and performs a secondary lighting correction. Instead of altering the original illumination context, FFE introduces an additional virtual light source so that the rendered face better matches

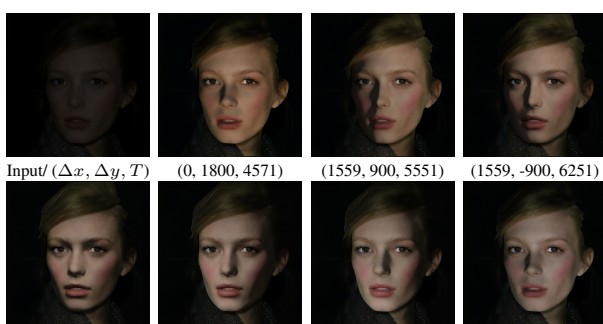

*Figure 1.* Controlling fill-light position and color temperature. Given the same input portrait, we move the virtual light along a circular trajectory while linearly increasing the color temperature, and keep the beam shape fixed. Each result is annotated with $(\Delta x, \Delta y, T)$, where $(\Delta x, \Delta y)$ denotes the lamp offset in pixels and $T$ is the corresponding color temperature in Kelvin.

aesthetic preferences or user-specified requirements. Conceptually, FFE simulates a virtual fill light with explicit control over its position, color temperature (Fig. 1), and other lighting attributes, lifting practical constraints of physical fill lights such as beam focus, power, and placement. In low-light and backlit portrait scenarios, image quality is often severely degraded by adverse illumination. Relying solely on an on-camera flash usually produces a strong point-light effect with harsh shadows and a fixed color temperature, which leads to visually unpleasant results. In such situations, photographers are typically advised to add external fill lights; however, these devices are often bulky and inconvenient to carry, which further motivates a virtual, deep-learning-based alternative.

Recent studies on face relighting have achieved promising results. Many existing methods (Hou et al., 2021; 2022) either explicitly include modules that suppress or normalize the original illumination, or implicitly encourage the model during training to become invariant to the lighting conditions in the input face image. As a consequence, the synthesized faces may no longer reflect the shadows, colors, and other cues implied by the current background illumination, which can lead to noticeable inconsistency between the foreground face and the background. A different line of work (Zhang et al., 2025b; Kim et al., 2024) aims to avoid such inconsistency by modifying the background together with the main subject, thereby changing the overall scene

---

[1]Shanghai Jiao Tong University. Correspondence to: [†]Yulun Zhang <yulun100@gmail.com>.

*Proceedings of the 43rd International Conference on Machine Learning*, Seoul, South Korea. PMLR 306, 2026. Copyright 2026 by the author(s).

illumination in a coherent way. However, both directions typically overlook another practical scenario: preserving the original background and its illumination while adding an extra light source that enhances the visual appearance of the face, which is exactly the setting addressed by FFE.

A major line of work (Zhang et al., 2025a) in portrait relighting estimates a 3D face representation, such as a mesh or neural 3D representation (Mildenhall et al., 2020; Kerbl et al., 2023), and then re-renders a 2D image under target illumination. When the 3D representation is sufficiently accurate, these approaches can produce realistic lighting and shadows. However, reconstructing high-quality 3D representations and obtaining consistent 2D renderings from a single image remains non-trivial. In parallel, complementary methods (Hou et al., 2022; Pandey et al., 2021) perform relighting directly in the 2D image domain without constructing a 3D representation. These methods leverage face priors or multi-illumination portrait data, enabling the network to learn facial 3D structure. Operating on the original image, they achieve photorealistic results for the given viewpoint by modeling and refining facial appearance in image space rather than reconstructing fully detailed 3D geometry.

To learn FFE, it is important to construct a multi-illumination dataset that covers a range of lighting conditions. Real facial data captured with light-stage systems (Debevec et al., 2000), such as Multi-PIE (Gross et al., 2010), provide variations in illumination and have facilitated the analysis of lighting effects. However, such systems are expensive and operate with a fixed or limited background, which restricts scalability and scene diversity. In parallel, several recent methods (Ponglertnapakorn et al., 2023; Han et al., 2023) rely on large-scale, in-the-wild face datasets and inject illumination priors through pretrained models. Their networks are trained to reconstruct the images, internalizing priors in facial illumination. Nevertheless, popular datasets such as FFHQ (Karras et al., 2019) deliberately restrict lighting to normal conditions, resulting in few examples with extreme illumination. In contrast, generating images by re-rendering faces under different lighting conditions makes it possible to construct more diverse illumination pairs with high fidelity. This also enables before-and-after lighting examples aligned with the FFE setting.

To address these issues, we construct LightYourFace-160K (LYF-160K), an FFE-oriented dataset with physically consistent illumination. Starting from a high-quality face dataset, we integrate real-world fill-light parameters into a custom rendering pipeline. During rendering, we parameterize the virtual fill light using six disentangled factors, such as color temperature, beam concentration, source-subject distance, and spatial position, and synthesize over 160K image pairs before and after fill lighting. Based on this pipeline and dataset, we first pretrain a physics-aware lighting-prompt

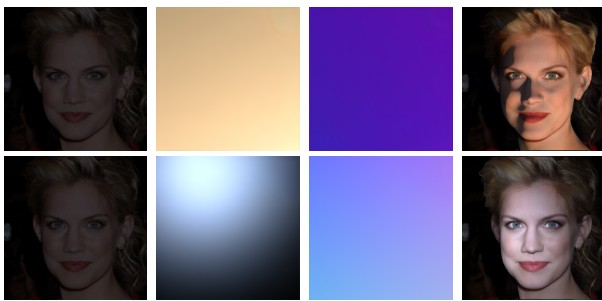

Input    Pred. Irradiance    Pred. Direction    FFE Image

*Figure 2.* Visualization of PALP predictions and effect on FFE. For two example inputs, we show the predicted planar irradiance map and direction field, together with the resulting FFE output generated under the corresponding lighting code.

(PALP) that regresses illumination conditions, thereby encoding physical priors into a compact representation. In the second training stage, we show that a diffusion model conditioned on this physically grounded lighting representation can faithfully interpret the 6D fill-light parameters as controllable lighting codes. This enables high-fidelity, user-controllable virtual fill lighting. Specifically, we fine-tune a pretrained diffusion backbone to obtain FiLitDiff, a one-step fill-light diffusion model conditioned on the PALP-produced lighting code, as shown in Fig. 2. After inference, we further enable training-free control of the fill-light strength using a wavelet-based adjustment scheme.

In summary, we make three key contributions:

- We build LightYourFace-160K (LYF-160K), a large-scale paired dataset for face fill-light enhancement, generated by a physically consistent renderer with a controllable 6D fill-light parameterization.

- We propose PALP, a physics-aware conditioning framework that injects lighting priors into diffusion models. PALP maps the 6D fill-light parameters to a compact lighting code that the denoiser can consume.

- With PALP conditioning, we develop FiLitDiff, a one-step diffusion model for high-fidelity, controllable virtual fill lighting with low computational cost, achieving strong perceptual quality on the held-out test set.

## 2. Related Work

### 2.1. Relighting

Recent studies on portrait relighting have achieved promising results. One line of work performs relighting in the 2D image domain, leveraging face priors or multi-illumination portrait data to learn lighting-to-appearance mapping (Hou et al., 2021; 2022; Pandey et al., 2021). Another line estimates a 3D face representation, such as a mesh or neural 3D representation, and re-renders the portrait under target illumination (Zhang et al., 2025a; Jiang et al., 2023). Beyond face-only relighting, several methods modify the back-

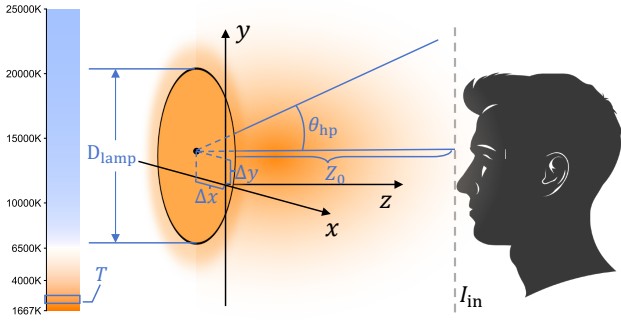

*Figure 3.* 6D control of our disk-shaped area fill light, including color temperature $T$, half-peak angle $\theta_{\mathrm{hp}}$, light-to-subject distance $Z_0$, disk diameter $\mathrm{D}_{\mathrm{lamp}}$, and image-plane offset $(\Delta x, \Delta y)$.

ground together with the subject, producing coherent scene illumination changes (Zhang et al., 2025b; Kim et al., 2024; Mei et al., 2024). Comprehensive Relighting extends this direction to general human relighting and harmonization in images and videos (Wang et al., 2025). This direction relates to background-aware portrait editing and scene-level illumination harmonization.

### 2.2. Diffusion-based Relighting

Diffusion models have recently been explored for portrait relighting, benefiting from strong generative priors and flexible conditioning interfaces. IC-Light imposes a physically motivated, consistent light transport constraint during training, encouraging illumination edits while preserving intrinsic appearance details (Zhang et al., 2025b). DiFaReli leverages conditional DDIM (Song et al., 2021) to decode a disentangled light encoding inferred from off-the-shelf estimators, and uses a rendered shading reference as spatial conditioning to facilitate modeling light-geometry interactions (Ponglertnapakorn et al., 2023).

### 2.3. Data for Relighting

Controlled light-stage capture systems provide systematic illumination variations and have facilitated relighting research (Debevec et al., 2000; Gross et al., 2010). In parallel, several recent methods rely on large-scale in-the-wild face datasets and inject illumination priors through pretrained models, training networks to reconstruct training images and internalize strong facial illumination priors (Ponglertnapakorn et al., 2023; Han et al., 2023; Ponglertnapakorn et al., 2025). Widely used high-quality face datasets such as FFHQ further serve as a common source domain in this line of research (Karras et al., 2019).

Synthetic paired data can also be constructed by applying physically based relighting pipelines to in-the-wild portraits, such as DPR, which generates a large-scale portrait relighting dataset with known SH lighting for controllable lighting supervision (Zhou et al., 2019). Recent volumetric portrait relighting systems are trained with light-stage captures and further improve training data quality via dedicated data-rendering strategies (Mei et al., 2024).

## 3. Method

### 3.1. Dataset Pipeline

During dataset construction, we take face images as input and combine their estimated depth, surface normal field, albedo, and specular coefficient to drive a physically motivated fill-light renderer that produces an additive illumination residual $\Delta I_{\mathrm{lamp}}$. The final relighting effect is controlled by a compact 6D lighting parameterization (Fig. 3), including the light color temperature $T$, the half-peak angle $\theta_{\mathrm{hp}}$, the light-to-subject distance $Z_0$, the light diameter $\mathrm{D}_{\mathrm{lamp}}$, and the light offset $(\Delta x, \Delta y)$ with respect to the image center. Specifically, for each input image $I_{\mathrm{orig}}$, we use Sapiens (Khirodkar et al., 2024) to predict the depth map $\mathbf{D}$ and normal map $\mathbf{N}$. We then segment the facial region to obtain a face-region mask $\mathbf{M_f}$, and feed the masked input into IntrinsicAnything (Chen et al., 2024b) to estimate the albedo $\boldsymbol{\rho}$ and the specular component $\boldsymbol{\beta}$. These together form the foundational geometric and material representations for physically consistent FFE rendering (see Fig. 4).

To make the "softness" of illumination controllable, our fill-light renderer adopts an area-light formulation with variable source size, where the fill light is abstracted as an emissive disk placed on a reference plane. To reduce the computational burden of evaluating the continuous area integral, we discretize the disk using a Fibonacci-spiral sampling strategy that generates an approximately uniform set of emissive points $N$. Concretely, we construct a deterministic point set by assigning each sample $k$ a radial coordinate and an angular coordinate according to:

$$r_k = \frac{\mathrm{D}_{\mathrm{lamp}}}{2}\sqrt{\frac{k+0.5}{N}}, \ \theta_k = k\pi(3-\sqrt{5}), \quad (1)$$

and obtain the 2D coordinates on the disk by $(r_k\cos\theta_k, r_k\sin\theta_k)$, for $k = 0, 1, \ldots, N-1$.

For a shading pixel $s$, its screen-plane coordinates together with the estimated depth determine its 3D location. Given the disk center offset $(\Delta x, \Delta y)$ and the light-to-subject distance parameter $Z_0$, the vector from the pixel to the $k$-th emissive point can be written as:

$$\mathbf{v}_{s,k} = \Big(\Delta x + x_k - x_s, \Delta y + y_k - y_s, Z_0 + \mathbf{D}_s\Big), \quad (2)$$

which yields the incident direction $\boldsymbol{\ell}_{s,k} = \mathbf{v}_{s,k}/\|\mathbf{v}_{s,k}\|_2$ and the distance $r_{s,k} = \|\mathbf{v}_{s,k}\|_2$. To model the directional emission of the fill light, in our renderer, we employ a cosine-lobe profile parameterized by the half-peak angle $\theta_{\mathrm{hp}}$. Specifically, with $\theta_{s,k} = \arccos(-\boldsymbol{\ell}_{s,k} \cdot (0,0,1))$ denoting the angle between the disk normal and the emission direction, the emission weight is calculated as:

$$w_{s,k}^{\mathrm{emit}} = \cos^p(\theta_{s,k}), \ p = \frac{\log(0.5)}{\log\big(\cos\theta_{\mathrm{hp}}\big)}, \quad (3)$$

so that the intensity drops to half at $\theta_{s,k} = \theta_{\mathrm{hp}}$, allowing explicit control over beam spread.

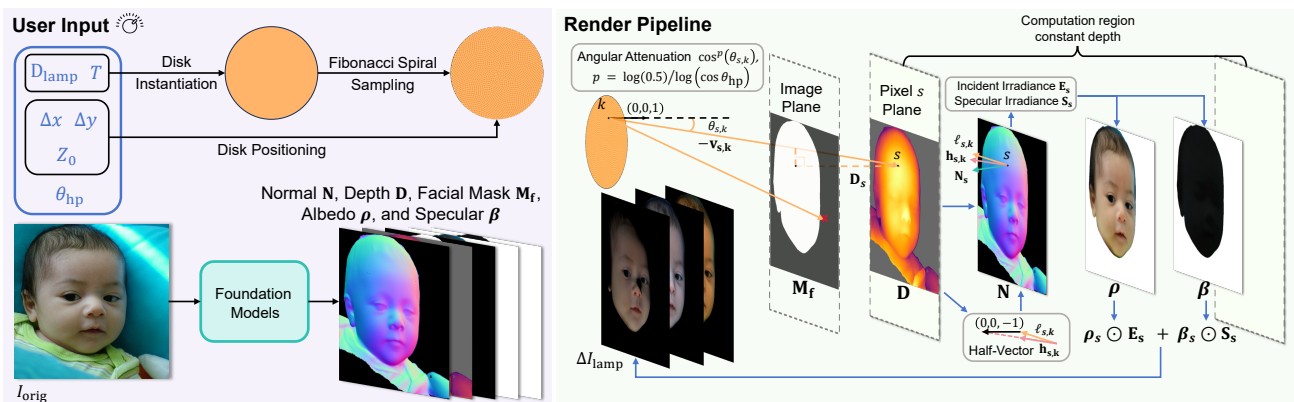

*Figure 4.* Overview of our physically consistent dataset pipeline and fill-light renderer. Given an input portrait $I_{\text{orig}}$, we estimate depth/normal, intrinsic albedo/specular, and a face mask, then render a disk-shaped area fill light controlled by 6D parameters to produce an additive residual $\Delta I_{\text{lamp}}$, yielding paired images before-after fill lighting.

The light color is parameterized by the correlated color temperature $T$, which is mapped to the CIE XYZ tristimulus values via a standard CCT-to-XYZ function $\mathbf{c}_{\text{XYZ}}(T)$ (Kang et al., 2002). In practice, we then modulate the per-pixel irradiance in XYZ space by the per-sample geometric gain and approximate the continuous disk integral using Monte Carlo averaging over $N$ samples:

$$\mathbf{E}_s^{\text{XYZ}} = \frac{1}{N} \sum_{k=1}^{N} \mathbf{c}_{\text{XYZ}}(T)\, w_{s,k}^{\text{emit}}\, V_{s,k}\, \frac{[\mathbf{N}_s^\top \boldsymbol{\ell}_{s,k}]_+}{r_{s,k}^2}, \quad (4)$$

where $[x]_+ = \max(x, 0)$, $\mathbf{N}_s$ denotes the surface normal at pixel $s$, and $V_{s,k} \in [0, 1]$ is the visibility term accounting for occlusion. Since we only have depth and no explicit geometric mesh, we estimate $V_{s,k}$ via a screen-space ray-marching scheme: we sample multiple points along the segment from the $k$-th emitter to pixel $s$ in image space, query the depth map at each sample location, and determine occlusion by comparing the ray depth with the sampled depth. To produce smoother penumbra boundaries, we replace the hard occlusion test with a continuous occlusion-probability accumulation and introduce mild step jitter to mitigate banding artifacts, resulting in a soft visibility mask. The resulting irradiance $\mathbf{E}_s^{\text{lin}}$ is converted to linear sRGB and used as the Lambertian diffuse incident term.

Moreover, we model specular highlights using a normalized Blinn–Phong model. With a fixed view direction $\mathbf{v} = (0, 0, -1)$, we define the half-vector $\mathbf{h}_{s,k} = (\boldsymbol{\ell}_{s,k} + \mathbf{v})/\|\boldsymbol{\ell}_{s,k} + \mathbf{v}\|_2$ and compute the normalized response:

$$S_{s,k} = \frac{n+2}{2\pi} \left[\mathbf{N}_s^\top \mathbf{h}_{s,k}\right]_+^n, \quad (5)$$

where $n$ denotes the shininess exponent. We aggregate the specular term over the $N$ disk samples in XYZ and transfer to linear RGB space, obtaining $S_s^{\text{lin}}$.

To perform energy computations in a linear color space, we convert both the albedo $\boldsymbol{\rho}$ and the specular component $\boldsymbol{\beta}$

from sRGB to linear RGB ($\boldsymbol{\rho}^{\text{lin}}$ and $\boldsymbol{\beta}^{\text{lin}}$). This process uses the standard inverse transfer function $I^{\text{lin}} = \phi\left(I^{\text{srgb}}\right)$, defined as (International Color Consortium, 2015):

$$C_{\text{lin}} = \begin{cases} \dfrac{C_{\text{srgb}}}{12.92}, & C_{\text{srgb}} \leq 0.04045, \\[2ex] \left(\dfrac{C_{\text{srgb}} + 0.055}{1.055}\right)^{2.4}, & \text{otherwise,} \end{cases} \quad (6)$$

which avoids luminance bias caused by additive operations in the non-linear sRGB domain. To prevent physically implausible energy amplification when the combined reflectance becomes overly large, we apply a per-pixel energy normalization on the reflectance maps in linear space: $\tilde{\boldsymbol{\rho}}_s^{\text{lin}} = \alpha_s\, \boldsymbol{\rho}_s^{\text{lin}}, \tilde{\boldsymbol{\beta}}_s^{\text{lin}} = \alpha_s\, \boldsymbol{\beta}_s^{\text{lin}}$. With a small constant $\varepsilon$ for numerical stability, the scale factor $\alpha_s$ is calculated as:

$$\alpha_s = \min\left(1, \frac{1}{Y(\boldsymbol{\rho}_s^{\text{lin}}) + Y(\boldsymbol{\beta}_s^{\text{lin}}) + \varepsilon}\right), \quad (7)$$

where $Y(\cdot)$ denotes the luminance operator in linear RGB, $Y(\mathbf{x}) = 0.2126 x_r + 0.7152 x_g + 0.0722 x_b$. This caps the summed reflectance while preserving the relative ratio. The diffuse component is obtained by modulating the incident irradiance $\mathbf{E}_s^{\text{lin}}$ with the albedo, and the specular component is obtained by modulating the aggregated specular irradiance $\mathbf{S}_s^{\text{lin}}$ with the specular. All operations are restricted to the facial region using the binary mask $\mathbf{M_f} \in \{0, 1\}$, where $\mathbf{M_f}(s) = 1$ indicates facial pixels. The total fill-light contribution in linear RGB is shown as follows:

$$\Delta I_s^{\text{lin}} = \left(\tilde{\boldsymbol{\rho}}_s^{\text{lin}} \odot \mathbf{E}_s^{\text{lin}} + \tilde{\boldsymbol{\beta}}_s^{\text{lin}} \odot \mathbf{S}_s^{\text{lin}}\right) \odot \mathbf{M_f}(s), \quad (8)$$

where $\odot$ denotes element-wise multiplication. We render the residual for every pixel $s$ and obtain the full-resolution fill-light residual map $\Delta I_{\text{lamp}}^{\text{lin}}$. Finally, we convert the residual back to sRGB using the inverse transfer function:

$$\Delta I_{\text{lamp}}^{\text{srgb}} = \phi^{-1}\left(\Delta I_{\text{lamp}}^{\text{lin}}\right). \quad (9)$$

## 3.2. Physics-Aware Lighting-Prompt (PALP)

For the FFE task, we inject a compact set of six lighting control variables into a diffusion model by pretraining a physics-prior injection module. Rather than rendering facial relighting directly, we adopt a simplified planar setup to construct auxiliary supervision, which reduces geometric and material complexity and yields more stable training signals while preserving physics-consistent priors.

In the parameter encoder of PALP, we transform the 6D lighting variables into a diffusion-compatible conditioning token sequence that matches the text-conditioning interface of Stable Diffusion (SD) (Rombach et al., 2022). We first apply an affine normalization so that each parameter mainly falls within $[-1, 1]$, reducing unit mismatch and dynamic-range differences. The normalized parameters are then injected into a set of learnable template tokens via FiLM-style modulation (Perez et al., 2018):

$$\mathbf{t}'_i = \mathbf{t}_i \odot (1 + \boldsymbol{\gamma}) + \boldsymbol{\sigma}, \qquad (10)$$

where $(\boldsymbol{\gamma}, \boldsymbol{\sigma})$ are predicted from the 6D input by an MLP. We further add sinusoidal positional encodings and apply a shallow Transformer encoder (Vaswani et al., 2017) to mix context along the sequence. It is followed by a linear projection to match the embedding dimension required by the diffusion model. The resulting token sequence is denoted as the conditioning prompt embedding $p$.

To make the tokens more identifiable and physically grounded, we introduce an auxiliary convolution-based token-to-image decoder during the pretraining stage. The branch mixes tokens along the sequence, reshapes them into a low-resolution feature map, and upsamples to predict planar lighting visualizations. Concretely, the decoder predicts two aligned outputs, a planar RGB irradiance map and a per-pixel direction field pointing from the disk center to each planar location, which are concatenated along the channel dimension. Together, these two targets constrain both the illumination magnitude and its spatial directionality in a computationally geometry-lightweight manner.

We train this module with online random sampling. At each iteration, we sample a 6D parameter vector $\{T, \theta_{\mathrm{hp}}, Z_0, D_{\mathrm{lamp}}, \Delta x, \Delta y\}$ from predefined physical ranges, with a small portion of long-tail perturbations. A physics-consistent renderer maps the sampled parameters to an RGB irradiance map $I_{\mathrm{plane}}$ and a per-pixel direction field $\mathbf{U}$. We concatenate them to form a 6-channel supervision target $\mathbf{Y} = \mathrm{Concat}\,(I_{\mathrm{plane}}, \mathbf{U}) \in \mathbb{R}^{H \times W \times 6}$. We then optimize the entire PALP end-to-end to predict $\hat{\mathbf{Y}}$ from the sampled parameters via the conditioning tokens, using a pixel-wise $\ell_1$ reconstruction loss:

$$\mathcal{L}_{\mathrm{pre}} = \left\| \hat{\mathbf{Y}} - \mathbf{Y} \right\|_1. \qquad (11)$$

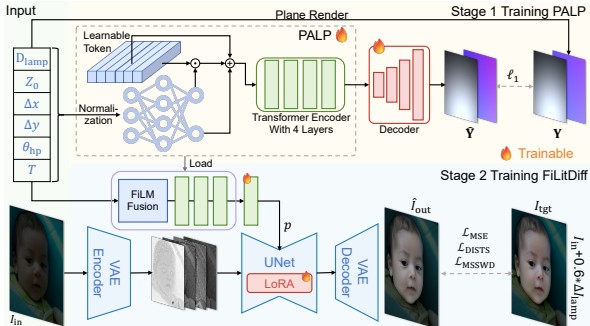

*Figure 5.* Overview of our framework. PALP encodes the 6D fill-light parameters into diffusion-compatible conditioning tokens via FiLM modulation and a shallow Transformer, and is pretrained with an auxiliary planar-light reconstruction decoder. The resulting lighting prompt conditions FiLitDiff, a one-step diffusion model fine-tuned from Stable Diffusion to perform controllable FFE.

## 3.3. Diffusion Model for FFE

To enable efficient one-step fill-light diffusion, we fine-tune a pretrained multi-step Stable Diffusion model into a single-step setting (Fig. 5). We first encode the input image $I_{\mathrm{orig}}$ into the latent space as $z_{\mathrm{in}} = E_\theta(I_{\mathrm{in}})$. We treat $z_{\mathrm{in}}$ as the latent state at scheduler step $t$. Following common image-to-image diffusion (Li et al., 2025; Gong et al., 2025) practice, we start from the latent of input image to preserve content that does not require modification. Compared with generation that relies on forward noising process, face fill-light enhancement (FFE) is closer to adding illumination cues on top of existing content. Therefore, we do not apply an explicit forward noising process during training. This choice makes the training objective closer to a conditional one-step refinement, where the model learns to inject illumination changes without perturbing background content. Given the lighting condition prompt $p$ and a fixed scheduler step $t$, the U-Net $\varepsilon_\theta$ predicts the noise term $z_\varepsilon = \varepsilon_\theta(z_{\mathrm{in}}; p, t)$.

We then reconstruct the clean latent using the deterministic DDIM update with $\sigma_t = 0$, which removes stochasticity and controls the noise magnitude. We first estimate the noise-free latent $z_0$ as follows:

$$\hat{z}_0 = \frac{z_{\mathrm{in}} - \sqrt{1 - \bar{\alpha}_t}\, z_\varepsilon}{\sqrt{\bar{\alpha}_t}}. \qquad (12)$$

The deterministic DDIM update can be written as:

$$z_{t-1} = \sqrt{\bar{\alpha}_{t-1}}\, \hat{z}_0 + \sqrt{1 - \bar{\alpha}_{t-1}}\, z_\varepsilon. \qquad (13)$$

In our one-step setting, the next step is $t - 1 = 0$, so $\bar{\alpha}_{t-1} = \bar{\alpha}_0 = 1$ and $z_{t-1} = \hat{z}_0$, which we denote as the clean latent $\hat{z}_{\mathrm{out}}$. We then decode it as the FFE output image $\hat{I}_{\mathrm{out}} = D_\theta(\hat{z}_{\mathrm{out}})$. After inference, we apply a training-free strength control based on the dual-tree complex wavelet transform (DT-CWT) (Selesnick et al., 2005). Because fill-light changes mainly affect low-frequency illumination, directly scaling the image residual may amplify high-frequency texture changes from VAE decoding

and generative denoising. We therefore modulate only the low-frequency wavelet subband, enabling continuous post hoc light-strength adjustment without rerunning diffusion. Specifically, for each color channel, we convert the input and output to linear RGB, apply DT-CWT, scale only the positive low-frequency residual $\Delta \mathbf{y}_+^{L,c} = [\mathbf{y}_{\text{out}}^{L,c} - \mathbf{y}_{\text{in}}^{L,c}]_+$, reuse the input high-frequency subbands, and convert the reconstruction back to sRGB:

$$(\mathbf{y}_s^{L,c}, \mathbf{y}_s^{H,c}) = \mathcal{W}\big(\phi(I_s^{\text{srgb},c})\big), \quad s \in \{\text{in}, \text{out}\},$$
$$I_{\text{mix}}^{\text{srgb},c} = \phi^{-1}\Big(\mathcal{W}^{-1}\Big(\mathbf{y}_{\text{in}}^{L,c} + x\Delta\mathbf{y}_+^{L,c}, \mathbf{y}_{\text{in}}^{H,c}\Big)\Big). \quad (14)$$

where $c \in \{R, G, B\}$, $\mathcal{W}$ and $\mathcal{W}^{-1}$ denote DT-CWT and inverse DT-CWT, $x$ is the user-defined strength, and $[\cdot]_+$ keeps the additive low-frequency fill-light residual.

The intermediate token sequence of PALP is set to match the dimensionality of SD prompt embeddings. Nevertheless, pretrained representations still exhibit a distributional gap from embeddings produced by the CLIP (Radford et al., 2021) text encoder in SD. To reduce this gap while avoiding prompt-representation degradation, we fine-tune the last Transformer layer of PALP and the linear projection. This adaptation preserves the physically meaningful structure learned in pretraining while producing conditioning prompts better aligned with the diffusion model interface.

**Training Objective.** For FFE, it is challenging to start from an input image that already contains non-ideal illumination and directly predict an additive lighting residual. This difficulty is further amplified because the VAE is not trained on the distribution of pure illumination residuals, making $\Delta I_{\text{lamp}}^{\text{srgb}}$ hard to reconstruct accurately. We therefore construct a carrier-based training target by randomly scaling the original image and the illumination residual. Specifically, we sample a scalar $\gamma \sim \mathcal{U}(0.2, 0.4)$ and form the target as:

$$I_{\text{tgt}} = \gamma I_{\text{orig}} + 0.6\,\Delta I_{\text{lamp}}^{\text{srgb}}. \quad (15)$$

We set the coefficient to 0.6 as a balanced target strength. Larger coefficients enhance the relighting effect and improve perceptual distance metrics, but may reduce image fidelity and destabilize color statistics. Thus, 0.6 provides a middle point for training, while the inference-time strength remains continuously adjustable through the wavelet control described above. This design uses $I_{\text{orig}}$ as a carrier to convey the additive lighting component through the VAE decoder. It also encourages the model to treat relighting as an incremental update. We supervise the output $\hat{I}_{\text{out}}$ using the $\ell_2$ loss and the DISTS loss (Ding et al., 2020). We additionally adopt MSSWD (He et al., 2024) as a color loss, since it is more sensitive to global color shifts induced by the color temperature. The overall objective is defined as:

$$\mathcal{L}_{\text{total}} = \mathcal{L}_{\text{MSE}}(\hat{I}_{\text{out}}, I_{\text{tgt}}) + \mathcal{L}_{\text{DISTS}}(\hat{I}_{\text{out}}, I_{\text{tgt}})$$
$$+ \alpha\,\mathcal{L}_{\text{MSSWD}}(\hat{I}_{\text{out}}, I_{\text{tgt}}). \quad (16)$$

# 4. Experiments

## 4.1. Experimental Settings

**Dataset Construction and Settings.** We construct LightYourFace-160K (LYF-160K) using the pipeline in Sec. 3.1. To make Monte Carlo averaging accurate and avoid high-frequency speckle artifacts caused by sparse emitter samples, we set the number of sampled points on the emissive disk to $N = 2048$. For parameter sampling, we adopt two strategies to improve lighting-condition coverage and generalization. First, for each input portrait, we generate three color-temperature variants (warm, white, and cool) and sample the color temperature from the corresponding range. Second, we use a long-tailed mixture distribution for the light offset $(\Delta x, \Delta y)$. This design better covers extreme fill-light positions and incident directions. Starting from 70,000 faces in FFHQ (Karras et al., 2019), we generate about 210,000 pairs and apply lightweight quality control by removing samples with failed masks or invalid lighting results. The rendering and filtering process costs 400 GPU-hours and yields 165,419 paired samples for training.

**Test and Validation Sets.** For evaluation, we follow the same pipeline on CelebA-Test (Karras et al., 2018) and additionally conduct manual verification, resulting in LYF-Val with 3,006 paired samples. Unlike simply applying a linear intensity scaling to the input images, we further process the validation inputs using Qwen-Image-Edit (Wu et al., 2025). Specifically, we edit half of the face images with an "underexposure" objective or a "side lighting" objective. This yields an additional 3,006 edited input pairs (denoted as LYF-EditVal), which we use to evaluate generalization beyond simple linear darkening. Additional results on the real-world captured dataset are provided in the supplementary material. This dataset is captured at night and provides better testing performance for nighttime conditions.

**Evaluation Metrics.** For our two paired validation datasets, LYF-Val and LYF-EditVal, we report both full-reference and no-reference metrics. For full-reference evaluation, we use PSNR, SSIM (Wang et al., 2004), DISTS (Ding et al., 2020), and LPIPS (Zhang et al., 2018). For no-reference evaluation, we use CLIPIQA (Wang et al., 2023), LIQE (Zhang et al., 2023), and TOPIQ (Chen et al., 2024a), using their face-oriented image quality assessment (IQA) variants.

**Implementation Details.** All training is conducted on a single NVIDIA RTX A6000 GPU. In stage 1, we train the PALP using AdamW (Loshchilov & Hutter, 2019) with a learning rate of $1 \times 10^{-4}$ and a batch size of 16 for 160k iterations. In stage 2, to balance reconstruction and color loss, we set $\alpha$ in Eq. (16) to $1 \times 10^{-2}$. We again use AdamW (Loshchilov & Hutter, 2019) with a learning rate of $1 \times 10^{-5}$ and a batch size of 2. The base model is SD2.1-base (Stability AI, 2022). We fine-tune the U-Net using LoRA (Hu et al., 2022) with rank 16 for 140k iterations.

| Methods | PSNR↑ | SSIM↑ | DISTS↓ | LPIPS↓ | MSSWD↓ | CLIPIQA↑ | LIQE↑ | TOPIQ↑ |
|---|---|---|---|---|---|---|---|---|
| Qwen-Image-Edit (Wu et al., 2025) | 13.48 | 0.7317 | 0.2126 | 0.3344 | 2.8623 | 0.6436 | 3.4799 | 0.7490 |
| DPR (Zhou et al., 2019) | 16.67 | 0.7198 | 0.1861 | 0.2448 | 1.8291 | 0.5957 | 2.8050 | 0.7365 |
| SMFR (Hou et al., 2021) | 12.02 | 0.4090 | 0.5208 | 0.5729 | 3.1989 | 0.2818 | 1.0257 | 0.1958 |
| IC-Light (Zhang et al., 2025b) | 7.83 | 0.5013 | 0.3476 | 0.5503 | 3.7624 | 0.6060 | 2.3073 | 0.6911 |
| FiLitDiff (Ours) | 25.65 | 0.8773 | 0.0756 | 0.0851 | 0.6284 | 0.6479 | 3.8122 | 0.7681 |

*Table 1.* Quantitative evaluation on LYF-Val. We report representative relighting baselines and a prompt-based editing reference that injects lighting instructions via prompts. The best results among relighting methods are highlighted in red.

| Methods | PSNR↑ | SSIM↑ | DISTS↓ | LPIPS↓ | MSSWD↓ | CLIPIQA↑ | LIQE↑ | TOPIQ↑ |
|---|---|---|---|---|---|---|---|---|
| Qwen-Image-Edit (Wu et al., 2025) | 13.26 | 0.6776 | 0.2300 | 0.3790 | 2.8918 | 0.6600 | 3.5911 | 0.7659 |
| DPR (Zhou et al., 2019) | 15.19 | 0.6181 | 0.2270 | 0.3227 | 2.1034 | 0.5307 | 2.5325 | 0.6606 |
| SMFR (Hou et al., 2021) | 11.91 | 0.4158 | 0.5176 | 0.5516 | 3.2311 | 0.2967 | 1.0496 | 0.2058 |
| IC-Light (Zhang et al., 2025b) | 7.27 | 0.4277 | 0.3811 | 0.6135 | 4.1194 | 0.5411 | 2.0373 | 0.6082 |
| FiLitDiff (Ours) | 24.01 | 0.8308 | 0.0982 | 0.1200 | 0.7438 | 0.6216 | 3.6649 | 0.7364 |

*Table 2.* Quantitative evaluation on LYF-EditVal. We report representative relighting baselines and a prompt-based editing reference that injects lighting instructions via prompts. The best results among relighting methods are highlighted in red.

| Physical Priors | | Metrics | | | |
|---|---|---|---|---|---|
| $I_{\text{plane}}$ | **U** | CLIPIQA↑ | LIQE↑ | DISTS↓ | SSIM↑ |
| × | × | 0.5839 | 3.7798 | 0.0763 | 0.8748 |
| ✓ | × | 0.5778 | 3.7560 | 0.0761 | 0.8731 |
| × | ✓ | 0.5839 | 3.7772 | 0.0772 | 0.8760 |
| ✓ | ✓ | **0.5849** | **3.8121** | **0.0756** | **0.8773** |

*Table 3.* Ablation on PALP pretraining supervision. We vary the physical-prior targets $I_{\text{plane}}$ and **U**, on LYF-Val. ×/× indicates training without PALP pretraining. Best results are in **bold**.

**Compared State-of-the-Art (SOTA) Methods.** We compare FiLitDiff with representative face relighting methods that cover complementary paradigms. Specifically, we include two learning-based face relighting methods, DPR (Zhou et al., 2019) and shadow-mask face relighting (SMFR) (Hou et al., 2021), as well as the diffusion-based relighting model IC-Light (Zhang et al., 2025b). Additionally, we report Qwen-Image-Edit (Wu et al., 2025) as a prompt-based editing reference to contextualize the FFE setting. For DPR and SMFR, we approximate our virtual fill light with spherical harmonics (SH). For IC-Light and Qwen-Image-Edit, we express the lighting parameters as textual prompts, using coarse tags with key parameters for IC-Light and more detailed descriptions for Qwen-Image-Edit.

### 4.2. Main Results

**Quantitative Comparisons.** Results on our validation sets constructed with the proposed pipeline (LYF-Val and LYF-EditVal) are reported in Tab. 1 and Tab. 2. We evaluate two input settings: uniformly downscaled inputs (LYF-Val), and inputs further edited to underexposure or side-light conditions using prompt-based editing (LYF-EditVal). Across both settings, FiLitDiff achieves the best performance among relighting baselines on full-reference and perceptual metrics, including PSNR, SSIM, DISTS, and LPIPS. It also attains the lowest MSSWD, indicating accurate color-temperature control. We additionally report Qwen-Image-Edit as a prompt-based editing reference; it attains competitive no-reference scores (*e.g.*, CLIPIQA, TOPIQ) but lags behind relighting methods on paired full-reference metrics under our physically parameterized evaluation.

**Qualitative Comparisons.** Comprehensive visual comparisons on the synthetic datasets are provided in Fig. 6 and Fig. 7, corresponding to LYF-Val and LYF-EditVal, respectively. Across both underexposure and side-lighting conditions, our method delivers improvements while maintaining high fidelity, whereas Qwen-Image-Edit and ICLight often introduce distortions. For underexposed inputs, whose lighting distribution is closer to our training data, FiLitDiff increases facial illumination and brightens the subject; compared with DPR and SMFR, which exhibit higher failure rates, our model demonstrates robustness and stability. For side-lighting cases, which are common in LYF-EditVal (Fig. 7), the results indicate generalization. These examples also highlight a practical role of FFE beyond brightening, namely improving shadow quality: under side lighting, part of the face can be strongly shadowed while the other part is over-bright, which is visually unappealing, and FiLitDiff enhances the shadowed side while preserving the original contrast pattern to improve appearance; in contrast, Qwen-style editing tends to disrupt the original contrast and cause a large shift in the perceived lighting condition.

**Ablation on PALP Pretraining Supervision.** The effect of PALP pretraining is studied in Tab. 3 by toggling the planar irradiance map $I_{\text{plane}}$ and the direction field **U**. Using both targets yields consistent improvements, indicating that the two cues are complementary. In particular, $I_{\text{plane}}$ mainly conveys the chromatic and intensity-related properties of the fill light, and provides supervision for parameters such as half-peak angle and disk size. However, without **U**, the model lacks guidance on the incident direction, mak-

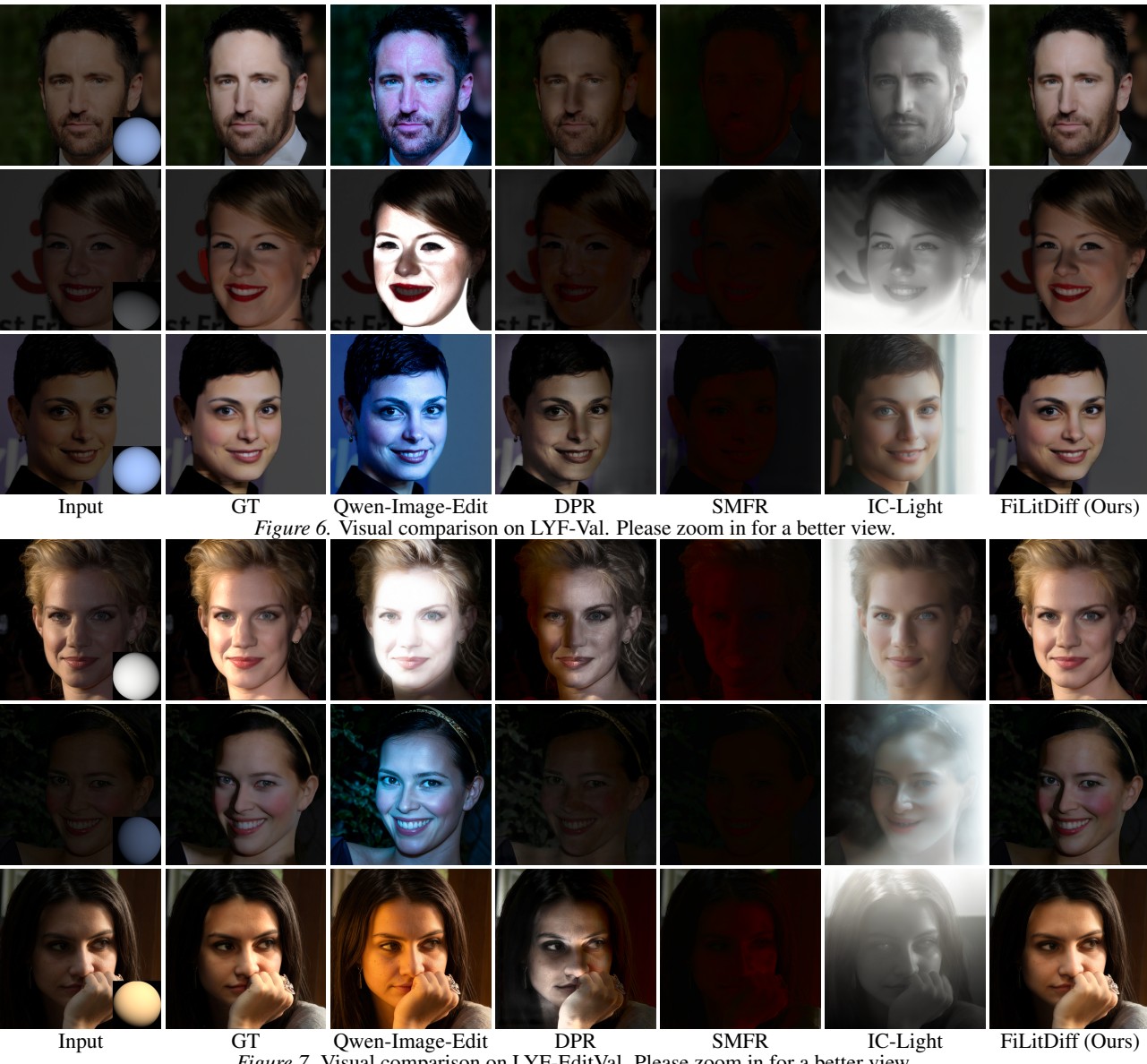

*Figure 6.* Visual comparison on LYF-Val. Please zoom in for a better view.

Input    GT    Qwen-Image-Edit    DPR    SMFR    IC-Light    FiLitDiff (Ours)

*Figure 7.* Visual comparison on LYF-EditVal. Please zoom in for a better view.

ing extreme lighting positions harder to disambiguate, and can reduce stability. Conversely, the only-**U** variant captures incident direction but lacks direct color and intensity supervision, so its conditioning signal remains incomplete. When PALP is trained jointly from scratch with the diffusion model, the supervision is entangled with facial geometry and appearance, so the module tends to absorb implicit illumination patterns present in the dataset rather than learning an explicit, parameter-aligned representation. Overall, the ablation supports pretraining PALP with paired irradiance and direction supervision to obtain a more physically grounded and reliable conditioning for FiLitDiff.

**Ablation on Training Objectives.** Using Tab. 4, we study the impact of different training losses. Using DISTS alone provides a reasonable training signal, indicating that a perceptual constraint helps preserve facial structure under fill-

| Training Losses | | | Metrics | | | |
|---|---|---|---|---|---|---|
| MSSWD | LPIPS | DISTS | C-IQA↑ | LIQE↑ | DISTS↓ | SSIM↑ |
| | | ✓ | 0.5835 | 3.7185 | 0.0769 | 0.8649 |
| ✓ | ✓ | | 0.5419 | 3.5842 | 0.0914 | 0.8703 |
| ✓ | | ✓ | **0.5849** | **3.8121** | **0.0756** | **0.8773** |

*Table 4.* Ablation on training objectives. Best results are in **bold**. In the table, C-IQA stands for CLIPIQA.

light changes. When we combine MSSWD with DISTS, the model becomes more faithful and stable, suggesting that color-aware supervision complements perceptual similarity by constraining global illumination and color shifts. In contrast, the MSSWD with LPIPS setting performs noticeably worse, implying that LPIPS is less aligned with our fill-light objective and may encourage unnecessary appearance changes. Overall, the results support training FiLitDiff with the complementary combination of MSSWD and DISTS.

## 5. Limitations

Our current study has several limitations. First, LYF-160K is built from public face data and does not intentionally filter by skin tone, ethnicity, or age. However, we have not conducted a controlled evaluation stratified by skin tone or other demographic attributes, so potential performance disparities remain unmeasured. Second, our data pipeline relies on external geometric and intrinsic predictors, whose structured errors may affect residual supervision, especially in extremely dark or low-texture regions such as hair, hairlines, and object boundaries. Finally, the current 6D disk-light parameterization supports a single fill-light source, leaving more complex lighting setups such as multiple lights, back lighting, and mixed lighting for future work.

## 6. Conclusion

This paper presents a physically consistent formulation for face fill-light enhancement (FFE) that emphasizes adding an explicit virtual fill light while preserving the original scene illumination and background. We introduce LightYourFace-160K (LYF-160K), a large-scale paired dataset synthesized by a renderer with a disk-shaped area light parameterized by six disentangled factors, enabling controllable and physically grounded supervision. Building on this data, we propose a physics-aware lighting prompt module (PALP) that converts 6D fill-light parameters into diffusion-compatible conditioning tokens via an auxiliary planar-light reconstruction objective. With PALP conditioning, we develop FiLitDiff, a one-step diffusion model fine-tuned from a pretrained diffusion backbone for fast and controllable FFE. Experiments on held-out paired validation sets demonstrate strong perceptual quality, competitive full-reference metrics, and improved preservation of background illumination. We hope this dataset and framework will facilitate future research on controllable, physically grounded portrait enhancement.

## Acknowledgments

This work is supported by the National Natural Science Foundation of China (62501386), CCF-Tencent Rhino-Bird Open Research Fund, and CAAI-Tencent Rhino-Bird Open Research Fund. This work is also sponsored by AI Hundred Schools Program and is carried out using the Ascend AI technology stack.

## Impact Statement

This work advances face fill-light enhancement, a task involving human face imagery. We use public datasets under their research terms and self-captured examples with consent. The method is intended for portrait enhancement and controllable relighting, not deceptive identity manipulation, surveillance, or unauthorized processing.

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
