# OpenReview forum: "Light Up Your Face: A Physically Consistent Dataset and Diffusion Model for Face Fill-Light Enhancement"
_ICML.cc/2026/Conference — ICML 2026 regular_

### Official Review · Reviewer_hz6Q · 2026-02-17

**Soundness:** 3
**Presentation:** 3
**Significance:** 3
**Originality:** 3
**Overall Recommendation:** 5
**Confidence:** 4

**Summary:**

The paper focuses on the problem of Face Fill-light Enhancement (FFE) where the goal is to artificially brighten the face in highly dark (underexposed) photos, by placing a virtual fill light with control over its position, orientation, intensity, color, etc. According to the authors, existing relighting methods focus on more general relighting settings and introduce inaccuracies when applied to the challenging FFE scenario. To address this task, the paper introduces a synthetic dataset of 160K source-target image pairs (underexposed - FFE enhanced) and uses it to finetune Stable Diffusion (SD) on the image translation task. To create the dataset, the authors use real-world images from FFHQ, and propose a sophisticated pipeline to simulate the pair of underexposed-enhanced settings, by using off-the-shelf estimators (deth, normals, albedo, etc) and a physically-based editing process. Each pair also comes with the 6 light parameters that represent the target fill light. The paper further proposes a trainable physics-aware encoder, named PALP, which transforms the 6D light condition to a token sequence suitable for SD conditioning interface via cross-attention. Then, they fine-tune SD using the conditioning embeddings and adapt it for one-step image-to-image translation in the FFE task.

**Compliance With Llm Reviewing Policy:**

Affirmed.

**Final Justification:**

The paper presents a solid contribution, including a proposed synthetic dataset for Face Fill-light Enhancement (FFE). The rebuttal offers additional ablation studies and visual evidence, and it addresses concerns regarding generalization to a broader range of subjects. Overall, I consider this a worthwhile contribution and recommend acceptance.

**Key Questions For Authors:**

1. Why does Qwen-Image-Edit achieve higher perceptual scores despite its unrealistic outputs?
2. How does the PALP module affect the results visually? Also, why is the case of using only the direction map U not included in the ablation?
3. In some cases (e.g. Fig. 6 last row, Fig.3 supp. 3rd row), the GT image looks unnatural with the face being much brigher than the hair. Is this a limitation of the data creation process?
4. Does the model generalize to video inputs? How consistent is the relighting result with dynamic facial expressions?

**Limitations:**

Limitations are not mentioned in the paper. I would suggest the authors to include them for completeness. For example, the proposed dataset is not designed for cases with multiple fill lights. While the experiments show decent performance in side-lighting scenarios, it is not clear how the model generalizes to more complex settings (e.g. back light).

**Strengths And Weaknesses:**

Strengths:

- Τhe paper is well-written and easy to follow. The method section is nicely structured, progressing from data creation, to lighting representation and diffusion model training with clear explanations and details in every section. The paper is overall of high quality.
- The proposed dataset will be a significant contribution for the community given the lack of similar "in-the-wild" datasets. Although the authors use Stable Diffusion, more recent image models (Flux, Qwen-Image) could also be used in combination with such a dataset.
- The paper shows convincing quantitative performance. It outperforms the baselines in both dark and side-lighting scenarios.
- The PALP conditioning framework is technically sound and experimentally valid. Although one would expect a simple injection of the 6D light parameters into the Unet (e.g. via an MLP) to work sufficiently, the ablation study of Table 3 indeed shows that the embeddings extracted by PALP learn more semantically meaningfull features that increase the generalization of the diffusion model. This is a decent contribution.
-    The model seems to handle real-world photos effectively (Fig. 2, supp.), which is the ultimate scope of the paper.

Weaknesses:
- It seems that the Qwen-Image-Edit baseline performs better in terms of no-reference metrics in side-lighting (Table 2, main) and real-world cases (Table 1, supp.). While the authors attribute this to the free-form generation of Qwen-Image-Edit which inflates these metrics, it is somewhat contradictory with the visual results of Qwen-Image-Edit shown in the figures which are heavily non-realistic and unexpected given the high perceptual score of this model. This raises some concerns regarding whether the images shown for Qwen-Image-Edit are representative of its quality or have been cherry-picked.
- The quantitative comparison in the supp. material uses 270 image samples created from 90 real-world portraits captured in low light settings. It would be nice to see more results from those 90 photos to check how the model performs in real settings. The supp. material provides only 3 visual examples (Fig. 2).
- I would like to see a visual ablation for the PALP module besides the quantitative study of Table 3. The paper lacks some failure cases showing the significance of pretraining PALP with paired irradiance and direction supervision.
- Missed reference: "Comprehensive Relighting: Generalizable and Consistent Monocular Human Relighting and Harmonization", CVPR 2025
- Minor spelling mistake in Fig.5: Dncoder -> Decoder

---

> ### Author Rebuttal · Authors · 2026-03-30
>
> # Response to Reviewer hz6Q (denoted as R4)
> `Q4-1:` Qwen-Image-Edit achieves higher no-reference perceptual scores in some settings, but its outputs look unrealistic. Why does this happen, and are the shown examples representative rather than cherry-picked?
>
> `A4-1:`
> Thank you for this question. To show the examples are **representative rather than cherry-picked**, we added Qwen-Image-Edit cases that receive high no-reference scores yet still look unrealistic on the [anonymous supplementary page][sp]. We believe this is because current no-reference IQA models are not fully adapted to face fill-light enhancement and may assign high scores to unrealistic editing artifacts. No-reference metrics are shown below.
>
> | Filename | CLIP-IQA↑ | LIQE↑ | TOPIQ↑ |
> |-|-|-|-|
> | 00001648 | 0.7115 | 4.9955 | 0.8250 |
> | 00002277 | 0.7822 | 4.8401 | 0.8387 |
>
> ---
> `Q4-2:` Only 3 real-world visual examples are shown for the 270 supplementary samples. Please provide more real-world examples and a visual PALP ablation.
>
> `A4-2:`
> Thank you for this suggestion. We have added more real-world examples and a visual PALP ablation to the [anonymous supplementary page][sp]. The ablation includes without $U$, without $I_{\\text{plane}}$, and without PALP pretraining, illustrating the effect of paired irradiance-and-direction supervision.
>
> ---
> `Q4-3:` Why is the “only $U$” setting not included in the PALP ablation?
>
> `A4-3:`
> Thank you for pointing this out. We originally did not include the “only $U$” setting because $U$ mainly encodes lighting direction, while $I_{\\text{plane}}$ provides the color and intensity cues that are central to face fill-light enhancement. Using only $U$ therefore yields a less informative and underdetermined conditioning signal, since it does not sufficiently encode the desired light magnitude and color.
>
> We agree that this ablation is helpful for a more complete analysis. We therefore report the quantitative results below and provide visual comparisons on the [anonymous supplementary page][sp].
>
> | Method | CLIPIQA↑ | LIQE↑ | DISTS↓ | SSIM↑ | MSSWD↓ |
> |-|-|-|-|-|-|
> | Only U | 0.5839 | 3.7772 | 0.0772 | 0.8760 | 0.6765 |
> | Ours | **0.5849** | **3.8121** | **0.0756** | **0.8773** | **0.6284** |
>
> ---
> `Q4-4:` In some cases, the ground-truth image looks unnatural, with the face much brighter than the hair. Is this a limitation of the data creation process?
>
> `A4-4:`
> Thank you for pointing this out. As also related to `A3-5`, this phenomenon is indeed one representative limitation of our current data creation pipeline. In very dark hair regions, the intrinsic decomposition step may underestimate the effective reflectance, making rendered lighting on hair overly weak. As a result, the generated target may occasionally show cases where the face appears much brighter than the hair.
>
> More fundamentally, hair is substantially harder to model than facial skin in our rendering-based pipeline. Unlike skin, hair has a much more complex appearance and light transport behavior, which is difficult to approximate well with a simple diffuse-plus-specular formulation. A possible direction for improvement is to handle hair illumination with a dedicated learning-based module. We will also discuss this issue in the limitations.
>
> ---
> `Q4-5:` Does the model generalize to video inputs? How consistent are the results under dynamic facial expressions?
>
> `A4-5:`
> Thank you for this question. We applied FiLitDiff to videos frame by frame and provide a GIF demo on the [anonymous supplementary page][sp]. The results suggest reasonable temporal continuity across adjacent frames, even under dynamic facial expressions, likely due to the one-step design and strong preservation of the original content and illumination structure. We observe slight flickering on smooth facial regions. Since FiLitDiff is not trained with temporal supervision, such artifacts remain. Temporal consistency could be further improved by introducing cross-frame consistency constraints.
>
> ---
> `Q4-6:` The paper does not clearly discuss limitations, such as multiple fill lights or more complex lighting settings like back lighting.
>
> `A4-6:`
> Thank you for this helpful suggestion. We agree that the limitations of the current work were not discussed clearly enough. Our current 6D disk-light parameterization is mainly designed for a single fill-light setting, so its generalization to more complex scenarios, such as multiple fill lights or back lighting is limited. Handling multiple fill lights would likely require modeling the composition of multiple lighting conditions, while back lighting may require a more flexible emitter parameterization. We will discuss this limitation in the revised paper.
>
> ---
> `Q4-7:` A relevant CVPR 2025 paper is missing from the references, and Fig. 5 contains a spelling error (`Dncoder` -> `Decoder`).
>
> `A4-7:`
> Thank you for pointing this out. We will correct both issues.
>
> [sp]: https://anonymous.4open.science/r/Submission_Number-6575-2812

---

> > ### Author Rebuttal · Reviewer_hz6Q · 2026-04-01
> >
> > The rebuttal addresses my concerns sufficiently. I recommend acceptance.

---

> > > ### Author Response · Authors · 2026-04-07
> > >
> > > Thank you for your careful reading and encouraging feedback. We sincerely appreciate your acknowledgement that our rebuttal has sufficiently addressed your concerns, especially through the additional real-world examples, PALP ablations, and supplementary visual evidence. We are very grateful for your time and consideration, and we especially appreciate your recommendation for acceptance.

---

### Official Review · Reviewer_L4BZ · 2026-03-01

**Soundness:** 3
**Presentation:** 3
**Significance:** 3
**Originality:** 3
**Overall Recommendation:** 4
**Confidence:** 2

**Summary:**

The paper introduces a novel framework for Face Fill-Light Enhancement (FFE), which focuses on adding a virtual fill light to underexposed or poorly lit portraits while preserving the original background and ambient illumination. To overcome the lack of diverse, extreme-lighting datasets, the authors construct LYF-160K, a large-scale dataset of 160K image pairs. In addition, the authors propose the Physics-Aware Lighting-Prompt (PALP) module, which is pretrained via an auxiliary planar-light reconstruction task to encode the 6D parameters into diffusion-compatible tokens. Finally, they present FiLitDiff, a one-step diffusion model fine-tuned from Stable Diffusion and conditioned on PALP, enabling fast, high-fidelity, and controllable FFE.

**Compliance With Llm Reviewing Policy:**

Affirmed.

**Key Questions For Authors:**

1. How does FiLitDiff generalize to real-world images beyond the supplementary nighttime dataset? For example, have you tested on diverse ethnicities, ages, or lighting extremes (e.g., mixed indoor-outdoor)?
2. Are other methods trained on the same data set? If not, is such a comparison fair?
3. What situations will the model not work well, and what are the failed cases?

**Limitations:**

The dataset involves a large number of face images and whether there are personal privacy issues?

**Strengths And Weaknesses:**

**Paper Strengths:**

1. The methodology is technically sound and well-engineered.
2. The paper is exceptionally well-written, easy to follow, and visually well-presented.
3. This method only needs one step diffusion to obtain high-quality results that conform to physical laws. It has significant practicability and computational efficiency, which makes it very attractive in practical applications.

**Weaknesses:**

1. The renderer relies on external predictors for depth/normal and intrinsic decomposition (albedo/specular) and on face masks. Errors and biases in these estimates can propagate into the synthetic supervision distribution, but the paper provides limited analysis of failure rates, error propagation, or robustness to such estimation noise.
2. The wavelet-based adjustment scheme is a training-free method, which I think is quite interesting, and it might be better if it could be introduced in the main text.

---

> ### Author Rebuttal · Authors · 2026-03-30
>
> # Response to Reviewer L4BZ (denoted as R3)
> `Q3-1:` The renderer relies on external predictors, but the paper provides limited analysis of failure cases, error propagation, and robustness.
>
> `A3-1:`
> Thank you for this important comment. As discussed in `A1-4` and `A2-2`, the external predictors are used as intermediate cues rather than error-free pseudo ground truth. The rendered fill-light residual is restricted to the facial region and stabilized by reflectance energy normalization and soft visibility modeling, which helps reduce the impact of estimation noise. We further remove samples with failed face segmentation and invalid renderings, refining 210K raw samples into 165K final training pairs. We will clarify failure cases and error propagation in the revision.
>
> ---
> `Q3-2:` The wavelet-based adjustment scheme is interesting and could be introduced in the main text.
>
> `A3-2:`
> Thank you for this suggestion. We agree that the wavelet-based strength control is an important practical component of FiLitDiff. In the revision, we will describe it more clearly in the main text, while keeping the technical details in the supplementary material.
>
> ---
> `Q3-3:` How does FiLitDiff generalize to real-world images beyond the supplementary nighttime dataset, such as diverse ethnicities, ages, or extreme lighting conditions?
>
> `A3-3:`
> Thank you for this question. We collected **90 real nighttime portraits** and generated three target-lighting variants for each image, resulting in **270 evaluation pairs**. We provide qualitative and quantitative results in the supplementary material, with additional examples on the [anonymous supplementary page](https://anonymous.4open.science/r/Submission_Number-6575-2812). These results support that FiLitDiff generalizes beyond synthetic and edited data and remains effective on real nighttime and low-light images. We have not yet established a systematic benchmark over ethnicity, age, or mixed/extreme lighting. In future work, we plan to build a broader real-world benchmark by curating more diverse portraits from public datasets and properly licensed public images, with screening for skin tone, age, and challenging lighting conditions such as mixed lighting and backlighting.
>
> ---
> `Q3-4:` Are the compared methods trained on the same dataset? If not, is the comparison fair?
>
> `A3-4:`
> Thank you for this important question. The compared methods are not trained on exactly the same dataset. We agree that the same-data setting would be fairer. To better address this concern, we take two additional steps. As discussed in `A2-4`, we first retrain DPR under the same local enhancement setting and report the corresponding results. Second, we supplement the evaluation with face-region metrics, which provide a fairer protocol for comparing global relighting methods with our local fill-light method.
>
> We align the test-time control interface to each model closely. Specifically, for DPR and SMFR, we approximate our 6D disk-light setting with spherical-harmonics lighting. For IC-Light and Qwen-Image-Edit, we convert the target lighting configuration into detailed textual prompts at inference time. Therefore, although this is not a fully controlled same-data comparison, we believe it still provides a meaningful evaluation against strong baselines.
>
> ---
> `Q3-5:` In what situations does the model fail, and what are the typical failure cases?
>
> `A3-5:`
> Thank you for this important question. Based on our current observations, the most representative challenging cases occur in extremely dark or low-texture regions, especially hair, the hairline, and boundary areas. In these regions, weak visual cues and less reliable target illumination make the learned fill-light response more prone to insufficient enhancement or locally unnatural transitions.
>
> These regions are intrinsically harder to model than facial skin, especially because hair has more complex appearance and light transport properties, and very dark regions further reduce the reliability of the estimated geometry and reflectance cues. We will clarify these failure cases in the revised version.
>
> ---
> `Q3-6:` Does the use of large-scale face datasets raise privacy concerns?
>
> `A3-6:`
> Thank you for raising this important concern. We agree that privacy is an important issue for face-related datasets and should be discussed explicitly.
>
> For the public data used in this work, FFHQ and CelebA-Test, we follow the original terms of use of the datasets and use them strictly for academic research. For the real images used in testing, the subjects provided informed consent for scientific research. For any real images shown in the manuscript, we also obtained explicit permission for academic publication and visualization.
>
> At the same time, we acknowledge that public availability or research licensing does not by itself eliminate privacy risks in face datasets. We will therefore add a clearer discussion of privacy and ethical considerations in the revised paper.

---

> > ### Author Rebuttal · Reviewer_L4BZ · 2026-04-01
> >
> > The author solved my problems and I keep my score.

---

> > > ### Author Response · Authors · 2026-04-07
> > >
> > > Thank you for your careful reading and positive feedback. We sincerely appreciate your acknowledgement that our rebuttal has addressed your concerns, especially regarding real-world generalization, fairness of comparison, failure cases, and privacy discussion. We are grateful for your time and consideration.

---

### Official Review · Reviewer_pFgZ · 2026-03-09

**Soundness:** 3
**Presentation:** 3
**Significance:** 3
**Originality:** 3
**Overall Recommendation:** 4
**Confidence:** 4

**Summary:**

This paper introduces LYF-160K, a physically consistent dataset for face fill-light enhancement (FFE). Building on this dataset, we propose FiLitDiff, a one-step diffusion model conditioned on a Physics-Aware Light Prompt (PALP) module. The model achieves realistic lighting effects while preserving the original scene illumination and background.

**Compliance With Llm Reviewing Policy:**

Affirmed.

**Final Justification:**

The author responses to my questions clearly. I keep my score.

**Key Questions For Authors:**

The paper presents a novel and well-motivated approach to Face Fill-Light Enhancement (FFE) alongside a robust, physically consistent dataset (LYF-160K). However, I recommend a borderline rating due to the lack of evaluation on challenging real-world datasets, potential unfairness in the current full-image quantitative metrics when comparing against global relighting baselines, and insufficient ablation studies on key methodological choices (e.g., the trade-offs of the one-step diffusion design and the fixed hyperparameter in Eq. (14)). For the rebuttal, I strongly suggest the authors provide qualitative and quantitative results on real-world photos, supplement the evaluation with face-mask-based regional metrics for a fairer comparison, and add the missing ablations or theoretical justifications for their core designs to better demonstrate the model's robustness.

**Limitations:**

See weakness.

**Strengths And Weaknesses:**

aper Strengths
	The paper identifies Face Fill-Light Enhancement (FFE) as a distinct subproblem within the portrait relighting domain. By prioritizing localized facial illumination optimization while strictly preserving the original ambient lighting and background integrity, this task formulation is both practically significant and clearly defined.
	The paper introduces the LYF-160K dataset, which innovatively integrates the intrinsic attribute decoupling capabilities of foundation models with 6D-parameterized physical rendering. By employing area light sampling and soft shadow computation, the dataset ensures the physical rigor of lighting effects while achieving large-scale generation. It provides a robust training baseline for the FFE task and serves as a valuable resource for future research in this community.
	The proposed FiLitDiff framework innovatively combines a Physical-Aware Light Prompt (PALP) module with a specialized one-step diffusion strategy. By employing an auxiliary pre-training task that decodes 6D physical parameters into planar irradiance and direction fields, the PALP module effectively grounds the latent tokens in physical light transport principles. Furthermore, by reconfiguring the diffusion process into a deterministic one-step mapping, the model achieves an excellent balance between high-fidelity facial enhancement and computational efficiency, making it highly suitable for practical applications.

Paper Weaknesses
Experiments
	Lack of evaluation on challenging real-world datasets. Although the paper conducts extensive quantitative evaluations on the LYF-Val and LYF-EditVal datasets, these images are essentially synthesized or edited using foundation models and image editing tools (e.g., Qwen-Image-Edit ). Consequently, there remains an inevitable domain gap between these datasets and real-world captured photos. Since the test sets do not fully cover highly challenging in-the-wild scenarios—such as extreme backlighting, complex ambient light interference, strong facial shadows, or occlusions—it is difficult to be fully convinced of the model's robustness in practical applications. It is highly recommended to include qualitative and quantitative comparisons on real-world datasets with extreme lighting conditions in the main text.
	Vulnerability to error propagation in the dataset generation pipeline. The construction of the LYF-160K dataset heavily relies on pre-trained foundation models (e.g., using Sapiens for depth/normal estimation and Intrinsic Anything for albedo/specular estimation ). If these prior models produce errors when handling complex portraits (e.g., inaccurate normal estimation due to occlusions or harsh original lighting), these errors will directly cascade and corrupt the generated pseudo ground truth. The paper currently does not detail any automated correction or robust filtering mechanisms to mitigate these propagated errors, beyond simply discarding failed segmentations. This could potentially limit the model's ability to learn truly accurate physically-consistent lighting.
	Insufficient ablation study and justification for the one-step diffusion design. The authors propose fine-tuning a multi-step diffusion model into a one-step model to significantly improve inference efficiency. However, there is a lack of comprehensive comparison with the standard multi-step diffusion process. Without such an analysis, it is difficult to assess the trade-off made by one-step inference regarding generative quality, particularly concerning the synthesis of high-frequency details and the realism of physical shadows.
	There may be fairness concerns regarding the quantitative evaluation: the current protocol directly compares baseline models designed for "global relighting" (e.g., IC-Light 、DPR) with our model, which focuses on "local relighting" while aiming to preserve the background, using full-image metrics (PSNR/SSIM). This comparison approach might impose unnecessary pixel-level penalties on the baseline models due to their intended background adjustments, leading to relatively lower scores and potentially overstating the advantages of our method. To ensure a more objective assessment, the authors are encouraged to consider supplementing the evaluation with face-mask-based regional metrics, or retraining the baseline models under the same local enhancement constraints to facilitate a fairer performance comparison.
Methodology
	Lack of theoretical or empirical justification for the hyperparameter in I_tgt=γI_orig+0.6ΔI_lamp^srgb (equation (14)). When constructing the training target in equation (14), the scaling coefficient for the fill-light residual termΔI_lamp^srgb  is hard-coded to 0.6. This setting appears rather empirical and arbitrary. The authors should provide a reasonable explanation or an ablation study demonstrating why a fixed value of 0.6 was chosen over other alternatives (e.g., dynamic sampling or simply 1.0). Investigating how this coefficient specifically affects the dynamic range of the final fill-light intensity and the convergence stability of the model would greatly enhance the methodological rigor of the paper.

---

> ### Author Rebuttal · Authors · 2026-03-30
>
> # Response to Reviewer pFgZ (denoted as R2)
> `Q2-1:` The evaluation remains mainly based on synthetic or edited data, with insufficient validation on challenging real-world photos.
>
> `A2-1:`
> Thank you for this question. We evaluate 90 real nighttime portraits with three sampled target fill-light settings per image, yielding 270 real-world evaluation pairs. Visual and quantitative results are provided in Supplementary Fig. 2, Supplementary Tab. 1, and the [anonymous supplementary page][sp]. These results support our method on real nighttime and low-light scenarios.
>
> ---
> `Q2-2:` The dataset construction relies heavily on foundation models such as Sapiens and Intrinsic Anything, yet the paper provides limited analysis of error propagation and robust filtering.
>
> `A2-2:`
> Thank you for this important comment. We do not treat the outputs of the foundation models as error-free pseudo ground truth. Instead, they are used as intermediate geometric and material cues to drive an explicit renderer that synthesizes an additive fill-light residual within the facial region. This design already helps limit how estimator noise propagates into the final supervision.
>
> We further apply targeted quality control during dataset construction. In particular, we remove samples with failed face segmentation, since **incorrect facial masks are a major cause of downstream estimation and rendering failures**. We also remove samples with invalid rendered illumination residuals. These filtering steps refines 210K raw candidates into 165K high-quality training pairs and improves the robustness of the supervision. However, we agree that such filtering cannot fully eliminate errors, and we will clarify this limitation in the revision.
>
> ---
> `Q2-3:` The paper lacks sufficient ablation or comparison for the one-step diffusion design. A more complete comparison with a standard multi-step diffusion process is needed to understand the trade-off between efficiency and generation quality.
>
> `A2-3:`
> Thank you for this suggestion. To assess the trade-off between efficiency and generation quality, we additionally trained a 10-step rectified-flow baseline under the same data and evaluation protocol. Results are shown below.
> | Method | CLIPIQA↑ | LIQE↑ | DISTS↓ | SSIM↑ | MSSWD↓ | Runtime (ms)↓ |
> |-|-|-|-|-|-|-|
> | 10 steps | **0.5906** | 3.5165 | 0.0957 | **0.8782** | **0.6122** | 361.47 |
> | Ours (1 step) | 0.5849 | **3.8121** | **0.0756** | 0.8773 | 0.6284 | **42.99** |
>
> The 10-step model is slightly better on some metrics, while ours is better on LIQE and DISTS and is about 8.4× faster. Visual comparisons are on the [anonymous supplementary page][sp].
>
> ---
> `Q2-4:` The current full-image evaluation may be unfair for comparing global relighting baselines with our local fill-light method. Please include face-region metrics or retrain baselines for a fairer comparison.
>
> `A2-4:`
> Thank you for pointing it out. We agree that full-image metrics may be unfair for this comparison. To make the comparison fairer, we retrained DPR under the fill-light setting and report face-region results below. **FiLitDiff remains best on face-region metrics.**
>
> | Method | DISTS ↓ | LPIPS ↓ | MSSWD ↓ | PSNR ↑ | SSIM ↑ |
> |-|-|-|-|-|-|
> | DPR | 0.2116 | 0.2440 | 1.8092 | 16.30 | 0.7575 |
> | DPR-retrain | 0.2173 | 0.2370 | 1.1697 | 18.45 | 0.7838 |
> | IC-Light | 0.2858 | 0.3786 | 2.4295 | 12.13 | 0.7128 |
> | Qwen-Image-Edit | 0.2175 | 0.2756 | 2.2042 | 14.58 | 0.7713 |
> | SMFR | 0.4867 | 0.4561 | 2.8195 | 12.31 | 0.5395 |
> | Ours | **0.1025** | **0.1055** | **0.7373** | **24.40** | **0.8601** |
> ---
> `Q2-5:` The fixed coefficient 0.6 in $I_{tgt} = \\gamma I_{orig} + 0.6 \\Delta I_{lamp}^{srgb}$ seems empirical and insufficiently justified. Please provide an explanation or an ablation study.
>
> `A2-5:`
> Thank you for this important question. This coefficient controls the trade-off between fill-light visibility and fidelity to the input. If it is too small, the fill-light effect becomes too weak for the model to learn a stable mapping from the 6D lighting parameters. If it is too large, image fidelity may be weakened. We also keep the coefficients of $\\Delta I_{lamp}^{srgb}$ and $I_{orig}$ no greater than 1 to avoid clipping, which could otherwise distort the correspondence between the control parameters and the resulting fill-light effect. We ablate coefficients 0.4, 0.6, and 0.8:
>
> | Coefficient | DISTS↓ | LPIPS↓ | MSSWD↓ | PSNR↑ | SSIM↑ |
> |-|-|-|-|-|-|
> | 0.4 | 0.1913 | 0.2512 | **0.8343** | **22.44** | 0.7781 |
> | 0.6 | 0.1744 | 0.2192 | 1.0067 | 21.69 | **0.7898** |
> | 0.8 | **0.1667** | **0.2108** | 1.2473 | 20.13 | 0.7820 |
>
> As the coefficient increases, perceptual metrics improve while fidelity declines. Therefore 0.6 is set as a balanced default during training, while the final fill-light strength **remains continuously adjustable at inference time** via training-free wavelet-based control scheme (Supplementary Section A).
>
> [sp]: https://anonymous.4open.science/r/Submission_Number-6575-2812

---

> > ### Author Rebuttal · Reviewer_pFgZ · 2026-04-02
> >
> > The author responses to my questions clearly. I keep my score.

---

> > > ### Author Response · Authors · 2026-04-07
> > >
> > > Thank you for your careful reading and positive feedback. We sincerely appreciate your acknowledgement that our rebuttal has clearly addressed your questions, especially regarding real-world evaluation, fairer face-region comparison, and the additional analysis of our key design choices. We are grateful for your time and consideration.

---

### Official Review · Reviewer_GNBj · 2026-03-12

**Soundness:** 2
**Presentation:** 2
**Significance:** 2
**Originality:** 2
**Overall Recommendation:** 4
**Confidence:** 4

**Summary:**

The paper propses a large scale dataset for face fill-light enhancerment. Together with the dataset, the authors provide a distilled one-step diffusion model to do the face fill-light enhancement. The proposed diffusion model builds on several pre-trained inverse rendering models which raises the concerns on the claimed physically grounded nature,

**Compliance With Llm Reviewing Policy:**

Affirmed.

**Final Justification:**

I thank the authors for performing dedicated experiments on different backbones, which addressed my concerns; therefore, I raise my final score to weak accept.

**Key Questions For Authors:**

1. Throughout the paper, most of the input images look very dark and flat (with fewer lighting-related effects). Is this because the proposed model struggles to remove and re-render the lighting effects?

2. What will happen if different foundation models are used for inverse rendering / surface normal estimation? The employed inverse rendering method, Intrinsic Anything, is a diffusion-based method and may hallucinate the predictions. How did you handle this?

3. Only photos with light skin are shown in the paper. Is this because of data bias or a flaw in the model design?

**Limitations:**

No limitations are provided in the paper. The potential negative societal impact should include inequality, as the method can only be applied to certain ethnic groups.

**Strengths And Weaknesses:**

Strengths

- A large-scale dataset.

- The 6D control of a disk-shaped fill light is physically grounded and meaningful.

Weaknesses

- Results look very plausible and do not show a clear advantage, given the fact that the model has been trained on exclusive data.

- There is no image with a dark-skinned face, and there is no discussion of this.

---

> ### Author Rebuttal · Authors · 2026-03-30
>
> # Response to Reviewer GNBj (denoted as R1)
> `Q1-1:` Results look very plausible and do not show a clear advantage, given the fact that the model has been trained on exclusive data.
>
> `A1-1:`
> Thank you for this comment. We would like to clarify that LYF-160K is not built from privately collected raw data, but rendered from public-source face images through our physically consistent FFE-oriented pipeline. Both the dataset and the model are intended to be released. Moreover, our evaluation is conducted on held-out test sets, namely LYF-Val and LYF-EditVal built from CelebA-Test.
>
> More importantly, our goal is not generic portrait relighting, but face fill-light enhancement, namely adding a controllable virtual fill light while preserving the original scene illumination and background. Under this task-specific setting, FiLitDiff shows clear advantages on both LYF-Val and LYF-EditVal in quantitative metrics and visual quality. We also provide fairer comparisons in the rebuttal, including face-region evaluation and retraining DPR under the same local enhancement setting (see `A2-4` and `A3-4`), where FiLitDiff still remains superior.
>
> Therefore, we believe our contribution lies not merely in training on a dedicated dataset, but in providing a complete FFE-oriented solution with task-aligned data construction, physically grounded conditioning, and efficient one-step inference.
>
> ---
> `Q1-2:` The paper only shows light-skinned faces and does not discuss performance across skin tones, raising concerns about data bias, cross-ethnicity applicability, and societal impact.
>
> `A1-2:`
> Thank you for raising this important concern. Our training set is constructed from FFHQ, which contains diverse subjects in age, ethnicity, and appearance, and we did not apply any filtering or bias toward specific skin tones during training. Therefore, the proposed method is not limited to light-skinned subjects.
>
> At test time, we use CelebA-Test, which also contains diverse facial attributes. We further provide qualitative examples on darker-skinned subjects in Figs. 3 and 4 of the supplementary material and on the [anonymous supplementary page](https://anonymous.4open.science/r/Submission_Number-6575-2812). These results indicate that the method can also work on darker-skinned subjects, although we agree that a systematic fairness evaluation is still needed.
>
> We also agree that this issue is under-discussed. In the revision, we will explicitly discuss **fairness concerns and potential performance disparities across skin tones and ethnic groups** in the limitations section.
>
> ---
> `Q1-3:` Throughout the paper, most of the input images look very dark and flat (with fewer lighting-related effects). Is this because the proposed model struggles to remove and re-render the lighting effects?
>
> `A1-3:`
> Thank you for the question. Our goal is not to remove the original illumination and re-render the entire image from scratch, but to preserve the original lighting distribution as much as possible while adding user-controlled fill light.
>
> For dark and visually flat inputs, such as some examples in Fig. 6, the model enhances the face according to the specified fill-light condition. For side-light cases, such as some examples in Fig. 7 and Supplementary Fig. 4, the model can also improve the shadowed regions while largely preserving the original contrast relationship. Therefore, the dark appearance of some inputs mainly reflects the challenging lighting conditions targeted by our task, rather than an inability of the model to handle lighting effects.
>
> ---
> `Q1-4:` What happens if different foundation models are used for inverse rendering or normal estimation? Since IntrinsicAnything is diffusion-based and may hallucinate, how is this issue handled in your pipeline?
>
> `A1-4:`
> Thank you for this important question. If stronger or complementary foundation models become available, they can be substituted or combined in our data construction pipeline to further reduce estimation errors.
>
> We do not treat the outputs of IntrinsicAnything as ground truth. In our pipeline, it only provides intermediate material cues for the explicit renderer to generate an additive fill-light residual. The rendering is further constrained by the facial region and stabilized by reflectance energy normalization and soft visibility modeling, which improves robustness to estimation errors. Moreover, the supervision remains condition-aligned through explicitly specified 6D fill-light parameters, and large-scale paired supervision can partially reduce the impact of estimation uncertainty.
>
> At the same time, systematic bias or structured hallucination may still remain. We will clarify this limitation in the revised version and investigate different estimators in future work.

---

> > ### Author Rebuttal · Reviewer_GNBj · 2026-04-05
> >
> > I thank the authors for their rebuttal. I still have concerns about the use of Intrinsic-Anything as the backbone. The human face is simple, and using a generic generative-based method for intrinsic decomposition on human faces will likely introduce significantly more noise than domain-specific deterministic models. Without testing on different backbones, it is hard to say if the proposed method really used the decomposition information or just found a way to leverage the Intrinsic-Anything priors, which cannot prove the correctness of the proposed method.

---

> > > ### Author Response · Authors · 2026-04-07
> > >
> > > Thank you for the further follow-up. We would first like to clarify the role of IntrinsicAnything [1] in our framework. It is **only used in the offline data construction pipeline** to provide intermediate material cues, such as albedo and specular components, for the renderer. The rendered results are then used to construct the training targets for FiLitDiff. Therefore, FiLitDiff does not directly use IntrinsicAnything during training or inference.
> > >
> > > To further address this concern, we additionally conduct a **cross-estimator sensitivity study**. We construct a 6K paired subset and, under the same setting, rebuild the training data using multiple alternative intrinsic estimators and train a separate model for each estimator for 20K iterations. These methods include three non-diffusion estimators, namely SPLiT [2], Intrinsic Compositing [3], and Colorful Diffuse IID [4], as well as the diffusion-based IntrinsicAnything [1].
> > >
> > > We evaluate the resulting models on LYF-EditVal. Since different estimators produce different synthetic targets, we use no-reference metrics for this supplementary experiment. The results are shown in the table below. After replacing the estimator, the alternative methods do not show a uniform performance collapse.
> > >
> > > We also add dataset pipeline examples and model prediction examples to the [anonymous supplementary page](https://anonymous.4open.science/r/Submission_Number-6575-2812). The former show training samples constructed with different estimators, while the latter show the corresponding test outputs of models trained on those data. Although the training samples constructed with different intrinsic estimators show some visual differences, the resulting models still produce reasonable fill-light enhancement results consistent with the task goal.
> > >
> > > Based on both the quantitative results and the visual comparisons, we hope this supplementary study can directly address the reviewer’s concern about dependence on IntrinsicAnything and its specific priors. Specifically, after explicitly replacing IntrinsicAnything, FiLitDiff can still be trained stably and produce reasonable results under other estimators. This suggests that our method **does not uniquely rely on the specific priors of IntrinsicAnything**. Instead, it can use decomposition information provided by different estimators during training. We will further clarify this point in the revised version. We will also systematically study the influence of different estimators on data construction and final model performance in future work.
> > >
> > > | Estimator | CLIPIQA | LIQE | TOPIQ |
> > > |---|---:|---:|---:|
> > > | IntrinsicAnything | 0.6396 | 3.6699 | 0.7545 |
> > > | Intrinsic Compositing | 0.6529 | 3.5662 | 0.7808 |
> > > | SPLiT | 0.6412 | 3.6681 | 0.7645 |
> > > | Colorful Diffuse IID | 0.5864 | 3.3574 | 0.7312 |
> > >
> > >
> > > [1] Chen et al., IntrinsicAnything: Learning Diffusion Priors for Inverse Rendering Under Unknown Illumination, ECCV 2024.
> > >
> > > [2] Fei et al., SPLiT: Single Portrait Lighting Estimation via a Tetrad of Face Intrinsics, TPAMI, 2024.
> > >
> > > [3] Careaga et al., Intrinsic Harmonization for Illumination-Aware Compositing, SIGGRAPH Asia 2023.
> > >
> > > [4] Careaga and Aksoy, Colorful Diffuse Intrinsic Image Decomposition in the Wild, ACM TOG, 2024.

---

### Review · Ethics_Reviewer_enHF · 2026-03-28

**Recommendation:** Remediation action needed

**Ethics Issue:**

The authors propose a dataset and model that has to do with people's faces, but the images in the dataset do not contain any dark-skinned faces. In order to ensure the model works on a broader variety of faces, I recommend that the authors include some dark-skinned faces in the dataset (and present some results with them).

---

### Decision · Program_Chairs · 2026-04-30

**Decision:**

Accept (regular)

**Comment:**

Overall, the paper presents a technically solid and well-engineered system with a clear task formulation and practical relevance. Multiple reviewers (e.g., hz6Q, pFgZ, L4BZ) acknowledge the sound methodology, the value of the proposed dataset, and the effectiveness of the one-step diffusion design, with several concerns being resolved after rebuttal.  The overall feedback trends positive after discussion, with most reviewers leaning toward weak accept and recognizing the contribution’s practical significance and extensibility.